# Graphs, Constraints, and Search for the Abstraction and Reasoning Corpus

**Yudong Xu**
Department of Mechanical & Industrial Engineering, University of Toronto
wil.xu@mail.utoronto.ca

**Elias B. Khalil**
Department of Mechanical & Industrial Engineering, University of Toronto
Scale AI Research Chair in Data-Driven Algorithms for Modern Supply Chains
khalil@mie.utoronto.ca

**Scott Sanner**
Department of Mechanical & Industrial Engineering, University of Toronto
ssanner@mie.utoronto.ca

## Abstract

The Abstraction and Reasoning Corpus (ARC) aims at benchmarking the performance of general artificial intelligence algorithms. The ARC's focus on broad generalization and few-shot learning has made it impossible to solve using pure machine learning. A more promising approach has been to perform program synthesis within an appropriately designed Domain Specific Language (DSL). However, these too have seen limited success. We propose Abstract Reasoning with Graph Abstractions (ARGA), a new object-centric framework that first represents images using graphs and then performs a constraint-guided search for a correct program in a DSL that is based on the abstracted graph space. Early experiments demonstrate the promise of ARGA in tackling some of the complicated tasks of the ARC rather efficiently, producing programs that are correct and easy to understand.

## 1   Introduction

In an attempt to better measure the gap between machine and human learning, the Abstraction and Reasoning Corpus (ARC) was created by Chollet in 2019. The dataset is a collection of 1000 image-based reasoning tasks, where each task asks for an output image given an input. To "learn" a procedure that produces said output, each problem comes with 2–4 input-output image pairs as training instances; these training inputs are different from the actual test input, but can be solved by the same (unknown) procedure. Some example problems from the ARC are shown in Figure 1. A 3-month competition with over 900 teams was hosted on Kaggle with the goal of solving the ARC (Kaggle 2020). Despite the massive effort, the resulting solutions only achieved 20% accuracy on the hidden test set, at best. In fact, the first-place solution could not solve any of the three examples shown in Figure 1 despite their simplicity to a human.

Recognizing and representing objects, actions performed on them, and relationships between them makes up a large portion of human cognition core systems (Spelke and Kinzler 2007). The ARC embodies this notion in its tasks. In fact, Acquaviva et al. (2021) found that when humans attempt to solve ARC tasks through language, half of the phrases they use relate to object detection. Therefore,

Neuro Causal and Symbolic AI Workshop at the 36th Conference on Neural Information Processing Systems (NeurIPS 2022).

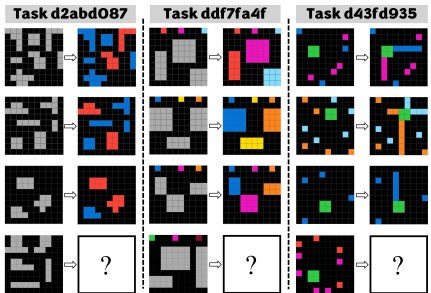

Figure 1: **Sample ARC Tasks.** Three tasks are shown. For a given task, each row contains one example input-output pair. The top three rows contain the "training" instances and the bottom row contains the "test" instance. The goal is to use the training instances to solve the test instance.

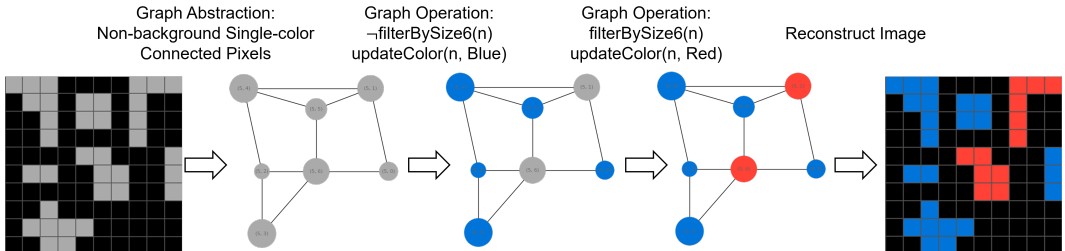

Figure 2: **Example solution by ARGA.** The input image is first abstracted into a graph in which each node represents a set of connected single-color non-background pixels. The solution colors in blue all nodes *not* containing exactly six pixels (size 6), then colors in red all nodes with size 6.

an object-centric approach to solving the ARC is highly promising. Surprisingly, this key insight is yet to be leveraged.

## 2 ARGA: Abstract Reasoning with Graph Abstractions

We propose a two-stage framework that takes an object-centric approach to solving an ARC task. First, the graph abstraction stage, where the 2D grid images are mapped to (multiple) undirected graph representations that capture information about the objects in the images at a higher abstracted level. Second, the solution synthesis stage, where a constraint-guided search is used to formulate the series of operations to be applied to the abstracted graphs that will lead to a solution. An example solution found by ARGA can be seen in Figure 2. The space of all possible operations are defined by a lifted relational Domain Specific Language (DSL) developed for ARGA.

Since the DSL defines operations on the abstracted graphs, this section will first formally define the graph abstraction stage. Then, the DSL will be defined in detail. Finally, the solution synthesis stage will be discussed.

### 2.1 Graph Abstraction

Graph abstraction allows us to search for a solution at a macroscopic level. In other words, we are modifying *groups of pixels* at once, instead of modifying each individual pixel separately. As a result, this approach has a smaller search space than its non-abstracted, raw image counterpart. The formal language we use builds on first-order logic which provides a flexible and expressive language for describing typed objects and relations. Object types in our DSL are shown in Table 1 and can be used as unary predicates, e.g., $Node(n)$ is true *iff* $n \in Node$. Relations between objects are shown in Table 2.

Let $i$ be any input or output 2D grid image from an ARC task. $i$ can be completely specified by its set of pixels $p$. Let $g$ be an abstracted graph with sets of abstracted nodes $n$. Each Node $n$ represents

$$Filter(x) ::= Type(x)$$
$$::= Filter(x) \wedge Filter(x)$$
$$::= Filter(x) \vee Filter(x)$$
$$::= \neg Filter(x)$$
$$::= \exists y\, Rel(x, y) \wedge Filter(y)$$
$$::= \exists y\, Rel(y, x) \wedge Filter(y)$$
$$::= \forall y\, Rel(x, y) \implies Filter(y)$$
$$::= \forall y\, Rel(y, x) \implies Filter(y)$$
$$::= Rel(x, c)\ [c \text{ is a constant}]$$
$$::= Rel(c, x)\ [c \text{ is a constant}]$$

$$Param(x, v) ::= v = c\,[c \text{ is a constant}]$$
$$::= Rel(x, v)$$
$$::= Rel(v, x)$$
$$::= \exists y\, Rel(x, y) \wedge Filter(y)$$
$$\wedge Param(y, v)$$
$$::= \exists y\, Rel(y, x) \wedge Filter(y)$$
$$\wedge Param(y, v)$$

Figure 3: **DSL Grammar**

an object that is detected in the original image $i$ based on the rules of the abstraction and relations between the nodes represent relationships between these objects.

Therefore, the graph abstraction process executes a mapping that generates some abstracted graph $g$ for image $i$. We note that there are multiple ways in which this mapping can be defined. Different graph abstractions can be used to identify objects in the image using different definitions of what an object is. An example abstraction is visualized in Figure 2.

| Object Type Set | Object Type Description |
|---|---|
| $i \in Image$ | A 2D grid image |
| $g \in Graph$ | An abstracted graph |
| $n \in Node$ | A node in an abstracted graph |
| $p \in Pixel$ | A pixel on an image |
| $c \in Color$ | Color (including *background*) |
| $s \in Size$ | Size of a node (# pixels) |
| $d \in Direction$ | Directions within the 2D image |
| $pa \in Pattern$ | A pattern found on the image |
| $t \in Type$ | Generic Type (any above) |

Table 1: **Object Types** in ARGA

## 2.2 A Graph DSL for the ARC

We now introduce a lifted relational DSL for ARGA built upon the objects and relations defined previously. The DSL is used to formally describe the filter language used to match node patterns, determine graph transformation parameters, and carry out transformations on abstracted graphs as described in the following.

| Typed Object Relations | Description |
|---|---|
| $containsNode(Graph, Node)$ | Graph contains Node |
| $containsPixel(Node, Pixel)$ | Node contains Pixel |
| $above(Node, Node)$ | Head Node is above tail Node in the 2D image |
| $below(Node, Node)$ | Head Node is below tail Node in the 2D image |
| $leftOf(Node, Node)$ | Head Node is left of tail Node in the 2D image |
| $rightOf(Node, Node)$ | Head Node is right of tail Node in the 2D image |
| $overlap(Node, Node)$ | Two Nodes are overlapping |
| $neighbor(Node, Node)$ | An edge exists between two Nodes |
| $color(Node, Color)$ | color of Node |
| $size(Node, Size)$ | size of Node |
| $Rel(Type, Type)$ | Generic Relation (any above) |

Table 2: **Object Relations** in ARGA.

**Filters:** Filters are used to select nodes from the graph. The fundamental grammar is a subset of first-order logic as shown in Figure 3 (Top). An example filter that matches nodes containing exactly 6 pixels is defined as $filterBySize6(n) \equiv Node(n) \wedge size(n, 6)$. We remark that $filterBySize6(n)$ and its negation ($\neg filterBySize6(n)$) are used to select nodes to be colored red and blue respectively in the example seen in Figure 2.

**Transformations:** Transformations are intuitively used to modify nodes selected by filters. They do so by modifying the values of object relations. An example transformation definition is as follows.

$$updateColor(n : Node, c : Color) \longrightarrow color(n, c) \wedge \neg color(n, c') \quad \forall c' \in Color \text{ s.t. } c' \neq c$$

In this example, the transformation *updateColor* updates ($\longrightarrow$) the color of the Node $n$ to $c$. It does so by assigning $color(n, c)$ to true and $color(n, c')$ to false for all other colors $c'$ in the abstracted graph representation. The full list of transformations and their descriptions can be found in the Appendix.

**Dynamic Parameter Transformations:** In the example shown in Figure 1 (Left), we can "statically" identify the color that the nodes should be updated to. However, this does not work for the example shown in Figure 1 (Middle), because the target color of a transformed grey object is that of its neighboring size-1 object. Therefore, we define parameter binding functions which allow us to dynamically generate parameters for transformations. The grammar for parameter binding is shown in Figure 3 (Bottom).

While sharing a similar grammar as filters, the $Param(x, v)$ has a special semantics different from filters that we pause to discuss. First, the goal of $Param(x, v)$ is to find possible matching parameters for an object $x$, hence we never apply a filter to $x$ in the grammar since we are not aiming to restrict it — $x$ is assumed given. Second, we can interpret $Param(x, v)$ as providing all possible parameter values $v$ that make $Param(x, v)$ true. However, we need a unique parameter $v$; if no $v$ matches for a given $x$ then $Param(x, v)$ fails to return a parameter and we cannot apply the transformation (it is considered a *noop*). If multiple $v$ match, then we deterministically order and return the first matching $v$. We remark that this dynamic parameter grammar includes static cases such as $Param(x, v) \equiv v = blue$, which would ignore the node $x$ and always return the parameter $blue$. Following is a more complex parameter binding:

$$bindSize1NeighborColor(x, v) \equiv \exists y \, neighbor(x, y) \wedge size(y, 1) \wedge Color(y, v))$$

Here, $bindSize1NeighborColor(x, v)$ matches (and returns) the color $v$ of any neighbor of $x$ with a size of 1 pixel. In the example shown in Figure 1 (Middle), suppose we have grey Node *n* selected by filters, we can then find the color to update it by calling $bindSize1NeighborColor(n, Color)$.

**Full Operation** With the filters, transformations and parameter bindings formally defined, we may now combine them to perform a full modification to the abstracted graph. Given a filter, a transformation, and $k$ parameter bindings $Param_i(x, v)$ ($i \in \{1 \ldots k\}$) for each parameter taken by the transformation (possibly none if $k = 0$), the full operation is shown in Figure 4.

for each $\quad n \in Node$
$\quad$ if $\quad filter(n)$
$\quad$ then $\quad v_i \hookleftarrow \{v | Param_i(n, v)\}$ for $i \in \{1 \ldots k\}$
$\qquad\qquad transform(n, v_1, \ldots, v_k)$

Figure 4: **Full Operation.** We assume that $\hookleftarrow$ deterministically selects a unique value $v_i$ if $|\{v\}| \neq 1$.

The set of operations required for solving the example in Figure 1 (Middle) are $filterByColorGrey$, $updateColor$ and $bindSize1NeighborColor$. We note that tasks such as the example shown in Figure 2 do not require dynamic parameters, in those instances, the parameter binding found in the solution simply returns a static value $v = c$.

### 2.3 Solution Synthesis

We implement a greedy best-first search, an illustration of which is shown in Figure 5. Suppose ARC task $t$ has $m$ training instances, with input-output images $\{input_i, output_i\}$ for $i \in \{1 \ldots m\}$. Each node in our search tree contains a set of graphs $\{g_{input\_i}\}$ for $i \in \{1 \ldots m\}$. $g_{input\_i}$ represents $input_i$ after the abstraction process and applying a set of operations $\{o_j\}$ for $j \in \{1 \ldots k\}$, where each $o_j$ is a full operation as defined previously. $k$ can be 0 which means no operations applied.

The primary metric used for node selection is how close it is to the target training output. For each node, we reconstruct the corresponding 2D image for each of the abstracted graphs $\{g_{input\_i}\}$. We then compare the reconstructed images with the training outputs $\{output_i\}$ and calculate a penalty score using Appendix Table 5. To expand a node with abstracted graphs $\{g_{input\_i}\}$, we first identify the set of all valid full operations $O$. Then, for each $o \in O$, we apply to $\{g_{input\_i}\}$ and obtain updated abstracted graphs $\{g'_i\}$ for $i \in \{1 \ldots m\}$. We add the new abstracted graphs $\{g'_i\}$ to the search tree as a new node and record the set of operations $\{o_j \text{ for } j \in \{1 \ldots k\}, o\}$ that led to it.

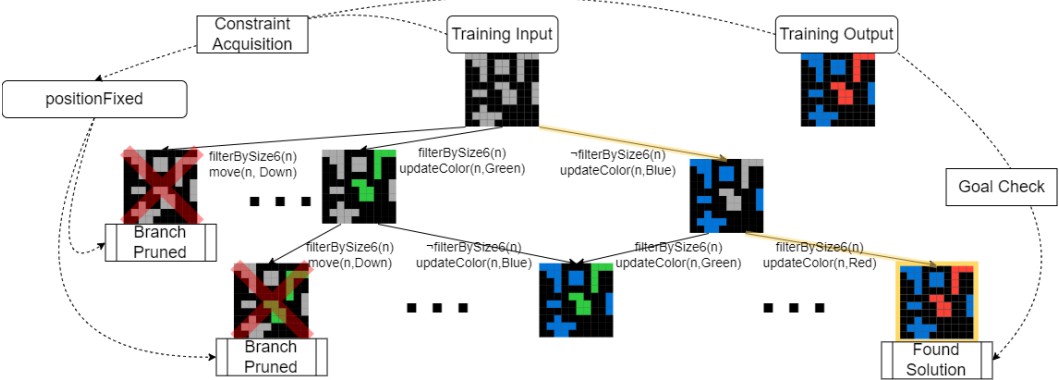

Figure 5: **Illustration of ARGA's constraint-guided search.** Note that a reconstructed 2D image is used at each node for better visualization, actual nodes consists a set of abstracted graphs.

**Constraint-Guided Search:** To reduce the size of the search space, we utilize constraints for pruning branches in the search tree. We illustrate this concept with an example. All objects in Figure 1 (Left) should not change in position. We can therefore define the constraint:

$$positionFixed(n : Node, n' : Node) \equiv \forall p \in P \; containsPixel(n, p) = containsPixel(n', p)$$

which returns True if a node and the updated version of that node share the same set of pixels, thus making sure that the node's position on the image remains fixed through the transformation. All transformations that modify a node's pixels can therefore be pruned by this constraint in the search tree. A visualization of pruning the search space with constraints is shown in Figure 5.

A set of constraints similar to the one shown above was defined for the ARC domain. To obtain the subset of constraints valid for pruning a specific search tree, we implemented a simple constraint acquisition algorithm influenced by the ModelSeeker (Beldiceanu and Simonis 2012) and Inductive Logic Programming (Lallouet et al. 2010).

While expanding a node in the search tree, we apply the same abstraction process for the output images $\{output_i\}$ as the input images to obtain $\{g_{output\_i}\}$ for $i \in \{1 \ldots m\}$. Then, for each full operation $o \in O$, we apply its filter operation $f$ to $g_{input\_i}$ and $g_{output\_i}$ to obtain pairs of $n_{in}$ and $n_{out}$. For each constraint $c$, if $c(n_{in}, n_{out})$ evaluates to True for all pairs found by $f$, all full operations $o$ with filter $f$ and transformation $t$ that violate constraint $c$ will be pruned.

## 3 Experiments

Chollet (2019) states that the ARC aims to evaluate "Developer-aware generalization", and all ARC tasks are unique and do not assume any knowledge other than the core priors. Therefore, implementing and evaluating ARGA on a subset of ARC tasks should provide useful insight into the effectiveness of our method without the need for extensive development of transformation functions, which are not the focus of our contribution.

Because it is our goal to focus on abstract transformations, we identified a subset of 160 object-centric tasks from the ARC and categorize them into three groups: (1) *Object Recoloring* tasks, which change colors of some objects in the input image. (2) *Object Movement* tasks, which change the position of some objects in the input image. (3) *Object Augmentation* tasks, which expand or add certain patterns to objects from the input images.

To further analyze the performance of ARGA, we evaluated the Kaggle Challenge's first-place solution (top quarks 2020) on the same subset of tasks. The code was executed with a search depth of 4. It is noted that due to the time constraint of the Kaggle competition, the code submitted for the competition used a mix of depth 4 and depth 3 across different tasks, which would result in slightly worse performance. We then recorded the results for the produced candidate with the highest score.

| Model | Task Type | # Training Correct | # Testing Correct | Average Nodes | Average Time (sec.) |
|---|---|---|---|---|---|
| ARGA | movement | 18/31 (58.06%) | 17/31 (54.84%) | 3830.35 | 89.75 |
| | recolor | 25/62 (40.32%) | 23/62 (37.10%) | 12316.87 | 326.83 |
| | augmentation | 20/67 (29.85%) | 17/67 (25.37%) | 4668.82 | 67.09 |
| | all | 63/160 (39.38%) | 57/160 (35.62%) | 7504.81 | 178.66 |
| Kaggle First Place | movement | 21/31 (67.74%) | 15/31 (48.39%) | 2176777.67 | 62.45 |
| | recolor | 23/62 (37.10%) | 28/62 (45.16%) | 2290441.32 | 93.19 |
| | augmentation | 35/67 (52.24%) | 21/67 (31.34%) | 2248151.10 | 66.07 |
| | all | 79/160 (49.38%) | 64/160 (40.00%) | 2249924.92 | 77.08 |

Table 3: **Results on subset of ARC.** *# Training correct* is the number of tasks that got all the training instances exactly right. *# Testing correct* is the number of tasks that got the testing instance exactly right. *Average Nodes* is the average number of unique nodes added to the search tree before finding a solution for correctly solved tasks. *Average Time (sec.)* is the average time in seconds to reach solution for correctly solved tasks.

## 3.1 Results

The resulting performance of ARGA as well as the Kaggle first place solution is shown in Table 3. With the exception of Object Movement tasks, our model performed slightly worse than the Kaggle first place solution in terms of accuracy. This is likely due to the solution space spanned by our DSL not being expressive enough, as it was developed using only a subset of the 160 tasks. On the other hand, the DSL used in the Kaggle solution was developed by first manually solving 200 tasks from the ARC (top quarks 2020).

Despite lower accuracy, ARGA achieves much better efficiency in search as we are able to reach the solution with 3 order magnitude less nodes explored. This suggests that with a more expressive DSL and a more efficient implementation (ARGA is currently implemented in python while Kaggle solution is implemented in C++), ARGA will be able to solve more tasks with less search effort.

Furthermore, the gap between the number of tasks solving all training instances and tasks solving the test instance is much smaller for ARGA. This suggests that ARGA is better at finding solutions which generalize correctly while the Kaggle solution often overfits to the training instances.

## 4 Conclusion

We proposed Abstract Reasoning with Graph Abstractions (ARGA), an object-centric framework that solves ARC tasks by first generating graph abstractions then performing a constraint-guided search. We evaluated our framework on an object-centric subset of the ARC dataset and obtained promising results. Notably, the efficiency in reaching the solution within the search space shows that with further development of the DSL, our method has the potential to solve far more complicated problems than state-of-the-art methods.

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

# A    Technical Details

## A.1    Full list of transformations

| Transformation | Description |
|---|---|
| updateColor(Node, Color) | Update color of Node to Color |
| move(Node, Direction) | Update pixels of Node to move 1 pixel in Direction |
| moveMax(Node, Direction) | Update pixels of Node to move in Direction until it collides with another node |
| rotate(Node) | Update pixels of Node to rotate it clockwise |
| fillRectangle(Node, Color) | Fill background nodes in rectangle enclosed by the node with Color |
| hollowRectangle(Node, Color) | Color all nodes in rectangle enclosed by the node with Color |
| addBorder(Node, Color) | Add additional pixels to Node in Direction |
| insertPattern(Node, Pattern) | Insert Pattern at Node |
| mirror(Node, Pixel, Direction) | Mirror Node toward Direction around Pixel |
| extend(Node, Direction) | Add additional pixels to Node in Direction |
| flip(Node, Direction) | Flip Node in place in some direction |
| transform(N, [k]) | Generic transformation with k parameters. |

Table 4: **Full List of Transformations**.

## A.2    Heuristic Function used in Search

| Actual | Predicted | Penalty |
|---|---|---|
| Background | Non-background | 2 |
| Non-background | Background | 2 |
| Non-background | Non-background wrong color | 1 |
| Non-background | Non-background right color | 0 |
| Background | Background | 0 |

Table 5: **Heuristic Function used in Search**

# B    Related Works

## B.1    Current ARC Solvers

There have been many attempts at solving the ARC. Most of those that have shown some success fall into the DSL within the program synthesis paradigm (Kaggle 2020). It has been shown that humans are able to compose a set of natural language instructions that are expressive enough to solve most of the ARC tasks, which suggests that the ARC is solvable with a powerful enough DSL and an efficient program synthesis algorithm (Johnson et al. 2021).

Indeed, this is the approach suggested by Chollet (2019) when introducing the dataset: "A hypothetical ARC solver may take the form of a program synthesis engine that uses the demonstration examples of a task to generate candidates that transform input grids into corresponding output grids."

Solutions using this approach include the winner of the Kaggle challenge, where the DSL was created by manually solving ARC tasks and the program synthesis algorithm is a search that utilizes directed acyclic graphs (DAG) to avoid duplicated search efforts. Each node in the DAG is an image, and edges between the nodes are transformations (top quarks 2020). The second-place solution introduces a preprocessing stage before following a similar brute-force search approach (de Miquel Bleier 2020). Many other Kaggle top performers shares this approach (Golubev 2020; Liukis 2020; Penrose 2020). Fischer et al. (2020) proposes a Grammatical Evolution algorithm to generate solutions within their DSL. Alford et al. (2021) utilizes an existing program synthesis system called DreamCoder(Ellis et al. 2020) to create abstractions from a simple DSL through the process of compression. The program then uses neural-guided synthesis to compose the solution for new tasks.

Other approaches to solving the ARC include the Neural Abstract Reasoner, which is a deep learning method that succeeds in a subset of ARC's problems (Kolev, Georgiev, and Penkov 2020). Assouel et al. (2022) developed a compositional imagination approach which generates unseen tasks for

better generalization. Ferré (2021) develops an approach based on descriptive grids. However, these approaches either were measured only on simplified version of the ARC or have shown limited success.

## B.2 Constraint Acquisition

Constraint Acquisition (CA) is a field that aims to generate Constraint Programming (CP) models from examples (De Raedt, Passerini, and Teso 2018). State of the art CA algorithms include Active CA (Bessiere et al. 2013; Arcangioli, Bessiere, and Lazaar 2016) requiring interaction from the user, and Passive CA (Bessiere et al. 2005) requiring only initial examples.

The passive CA algorithm used for ARGA was influenced by ModelSeeker (Beldiceanu and Simonis 2012), which finds relevant constraints from the global constraint catalog (Beldiceanu, Carlsson, and Rampon 2005) as well as the system developed by Lallouet et al. (2010) which uses Inductive Logic Programming (ILP) and formulates constraints from logical interpretations.

