# OpenReview forum: "Graphs, Constraints, and Search for the Abstraction and Reasoning Corpus"
_NeurIPS.cc/2022/Workshop/nCSI — nCSI WS @ NeurIPS 2022 Poster_

### Official Review · Reviewer_zjuG · 2022-10-08
**Good technical quality but the presentation can be improved**

**Rating:** 2
**Confidence:** 2

**Review:**

## Summary
The paper proposes a new framework to address the Abstraction and Reasoning Corpus (ARC). ARC is difficult to solve because the agent needs to know the complex relations of objects to find a solution. To mitigate the problem, the paper proposes Abstract Reasoning with Graph Abstractions (ARGA), which performs abstraction of the raw input to the form of graphs, and solves the problem by constraint-guided search. In the experiments, ARGA produced slightly worse accuracy compared to the kaggle 1st place method but achieved an efficient search, i.e., outperformed the benchmark in terms of the number of nodes to be expanded during the search.

## Pros
- The paper has a good-quality technical section
- The paper presents the method clearly
- The framework seems to be novel in terms of the composition of the object-centric approach with constraint-guided search
- The framework can lead important applications for tasks where the agent needs to perform problem-solving based on the objects and its relations

## Cons
- The paper lacks a (formal) description of the benchmark. I am not sure if ``Kaggle First Place" is a proper description as a benchmark to be compared with. A formal description of the benchmark would be needed, e.g., what kind of search algorithm is used, and how much computational cost (in terms of memory and time)  is required. Since the average time is shown in the table but the implementation setting (e.g. programming language) is different. So it is not easy to compare the efficiency in terms of computational time. A formal analysis of the algorithm in terms of computational complexity would help to show the efficiency of the proposed framework.
- Writing can be improved. Some minor points I noticed are listed:
  - The text and the edge label in Fig. 2 are too small to read
  - I believe function and variable names that have more than one character in the math mode should be written using a typeset, e.g., \mathit.
  - I found that $g_{in\\_i}$ is a bit confusing because $i$ and $n$ appear as variable, respectively, and $in$ and $i$ appear in the same font.

---

### Official Review · Reviewer_cMEF · 2022-10-14
**Interesting object-centric approach which is well-described and easy to understand but experiments seem to be weakly executed and not well discussed.**

**Rating:** 1
**Confidence:** 2

**Review:**

### Summary
The paper proposes an object-centric approach to tackle the tasks in the Abstraction and Reasoning Corpus. Their approach consists of two stages, parsing the task’s images to graph representations and a constraint-guided search to construct operations to solve the task.   In the experimental evaluation, the introduced approach is compared to a corresponding Kaggle Challenge’s first-place solution.


### Strength
The introduced object-centric approach is well-described and easy to understand.
The experimental evaluation is designed to analyze the object-centric tasks. To this end, a subset of the ARC was extracted and separated into different categories.


### Weaknesses
- However, I’m missing a deeper discussion on the benefit of an object-centric solution. The authors mention that ARGA is performing better than the baseline on object movement tasks but then focus on why it performs worse on the others.

The authors highlight the efficiency of their approach and state that their methods solution space may not be expressive enough.
Could a more expressive DSL lead to decreased efficiency?

Unfortunately, the evaluation lacks appropriate cross-validation.

Furthermore, related object-centric work is missing. Especially I would be interested in a discussion of other object-centric approaches which could be used to extend the introduced approach beyond the ARC.


### Minor
Typo Line 9: demonstrate
Typo Line 141: For each node, we
Typo Line 174: To further analyze the performance of ARGA, we

---

### Meta-Review · Program_Chairs · 2022-10-20

**Recommendation:** 2
**Confidence:** 3

**Metareview:**

Both reviewers agree that the approach is interesting. Generally, there is a consensus that the method is easily understood, even if one of the reviewer highlighted the need for improved presentation. On the contra side of things, one reviewer lacks the overall motivation to the proposed approach. However, as the other reviewer suggests the framework can lead to important applications for tasks where the agent needs to perform problem-solving based on the objects and its relations. Therefore, accept is recommend.

---

### Decision · Program_Chairs · 2022-10-20

Accept (Poster)